# Methylene Blue and Proflavine as Intraarterial Marker for Functional Perforazome—Comparative Study

**DOI:** 10.3390/jpm11020147

**Published:** 2021-02-19

**Authors:** Maria-Eliza Nedu, Mihaela Tertis, Cecilia Cristea, Alexandru Valentin Georgescu

**Affiliations:** 1Department of Plastic Surgery, Faculty of Medicine, Iuliu Haţieganu University of Medicine and Pharmacy, 400347 Cluj-Napoca, Romania; eliza.nedu@gmail.com (M.-E.N.); valentin.georgescu@umfcluj.ro (A.V.G.); 2Department of Analytical Chemistry, Faculty of Pharmacy, Iuliu Haţieganu University of Medicine and Pharmacy, 400349 Cluj-Napoca, Romania; mihaela.tertis@umfcluj.ro

**Keywords:** perforasome tracer, methylene blue, proflavine, in vivo study, sentinel lymph nodes

## Abstract

Methylene blue (MB) is both a dye and a medicine known and used for a long time including as lymphatic tracer in melanoma and breast cancer for revealing sentinel lymph nodes. Proflavine (PRO) is an acriflavine dye, used as bacteriostatic disinfectant against many gram-positive bacteria that was also successfully applied to evaluate morphopathological changes in tissues. This study was performed on a group of twenty-eight Wistar rats and had as its main objective the in vivo evaluation of the use of MB and PRO as perforator tracers. The two dyes proved to be effective functional perforasome tracers with medium inflammatory infiltrate in the skin of the island perforator flap which heals perfectly at 14 days with complete absence of the inflammatory reaction. At the same injected amount, PRO seems to determine a greater inflammatory reaction compared with MB, but in smaller concentration, the inflammatory response is absent in the case of PRO. In conclusion, both substances tested within this in vivo study are good functional perforasome tracers, but PRO has the advantage of the absence of inflammatory reaction when using lower concentrations, while preserving unalerted its efficiency as tracer.

## 1. Introduction

Methylene blue (MB) also known as methylthioninium chloride is both a dye and a medicine known and used for a long time. The use of MB as a tracer has been known for almost 30 years when Morton et al. [1] introduced it in 1992 as intraoperative lymphatic tracer in early-stage melanoma. A few years later, in 2001, Simmons et al. [2] proposed MB, as alternative to isosulfan blue, for identifying sentinel lymph nodes in T1 and T2 breast cancer patients [3] with 91%–93% efficiency [4,5]. The same researchers also used indocyanine green (ICG) [6,7] in combination with MB, the first being a well-known and often used dye in perforator studies [8].

Tanaka et al. [9] used MB for coronary arteriography and cardiac perfusion assessment, allowing direct visualization of the vessels. When intraarterially administrated, MB does not bind to proteins, and it has high tissue uptake and slow washout (40–60 min) [9].

Recently, ICG, a cyanine dye used in medical diagnostics, started to be used as perforator tracer. It was reported that ICG binds to plasmatic proteins and has a half-life of about 3–5 min in humans [10], and its concentration decreases rapidly in the first five minutes after administration, after which the decrease is slower, and stabilizes after 30 min [11]. One of the applications of the perfusion with dyes is the study and evaluation of perforators [12]. However, when the thickness of the flap is under 2 cm, the perfusion is useful for perforators’ mapping [11].

Proflavine (PRO) is a similar dye, whose concentration in the blood flow decreases rapidly in the first 3–5 min after administration [13,14]; then, the decrease is slower and stabilizes for around 30 min after the initial dosing [14]. Confocal fluorescence microscopy tests were done in the presence of PRO as stain for the evaluation of the breast [15], for the in situ evaluation of the changes in adipocytes in invasive ductal carcinoma and ductal carcinoma [16] and for the hyperspectral imaging assessment of tumour margin in head and neck cancer surgery [17,18]. The same dye was used for in vivo imaging in micro-endoscopic identification of breast, head and neck cancer (18), dysplastic regions of colorectal tissue [19], oral [20] and cervical epithelial cells [21], and it was observed that it had no influence on the evolution of the tumour despite its ability to intercalate into the DNA structure [22,23,24].

Despite the ability of MB and PRO to intercalate in the structure of DNA, no negative influences on tissues viability were observed. The structural formulas of the two compounds are shown in Figure 1.

Leung et al. [25] introduced the concept of functional perforasome based on more anatomic perforasome units. Its assessment can be done only in vivo, and the choke vessels which bind the anatomic perforasomes strongly influence the surface of functional perforasome and the number of irrigated anatomical units, respectively.

The aim of this study was to compare from a clinical and histopathological point of view the use of the two dyes: MB and PRO as functional perforasome tracers. In this study, levels of the concentration of infused dye solutions in rats were high enough to allow the direct visualization with the naked eye, namely: 0.65 mg/mL for MB and 1 mg/mL for PRO, respectively.

## 2. Materials and Methods

The study was conducted in the animal research laboratory, respecting all the regulations regarding animal studies, with the approval of the ethical committee certified with the authorization No. 199/07.02.2020. Twenty-eight male Wistar rats, weighting between 400–500 g, were used for this in vivo study.

All the animals were housed under standard environmental conditions, in rat cages and offered standard rat chow and tap water. They were housed individually in the animal facilities of the institution under controlled temperature and humidity, and they were photographed at different days post operatory. No prophylactic treatment with antibiotics was used for any of the animals in the study group. The rats were randomly assigned in two groups of two lots of 7 subjects, based on the type of dye injected. Standard solutions of the two fluorescent dyes of 10 mg/mL were prepared in 0.9% serum solution for infusions (B. Braun, Melsungen AG, Germany). These standard solutions were further diluted with serum to optimal concentrations for in vivo tests on rats, namely, 0.65 mg/mL for MB and 1 mg/mL for PRO. A series of solutions with concentrations between 0.16–1.6 mg/mL for MB, and 0.1–1 mg/mL for PRO, respectively, were prepared and injected into rats. The lowest concentration which allowed easy observation with the naked eye after injection was chosen for each dye. Equal volumes of 1 mL of diluted MB or PRO solutions were used for the subjects as mentioned above [26].

The dyes were injected in femoral artery after perforator selection and ligature of the rest of the branches of femoral artery and respective superficial epigastric artery. After direct identification of a functional perforasome (see the Results section of the manuscript for more details) a flap was raised based on it. The procedure is described on detail in a previous study [26].

### 2.1. Flap Assessment

The two groups of rats were monitored postoperatively, and their health status was assessed. Skin island specimens were harvested subsequently following the dye injection in days 1, 2, 3, 4 and 14 to assess the time-based evolution of all the structures of interest: femoral artery and vein, the superficial epigastric vein, the selected perforator vessels, and the tegument island.

### 2.2. Statistics

Statistical analysis was performed with Prism software (GraphPad Software, San Diego, CA, USA) evaluating the mean and standard error. For normally distributed data, comparison between the two groups was performed using student test, Mann–Whitney U Test. The result showed no significant differences between the two groups if *p* < 0.05.

### 2.3. Histological Evaluation

Tissue samples harvested from the island flaps were analysed. First at the 4th and then at the 14th post operatory day, independent samples per experimental group were analysed. Skin orientation (and cutting direction) was kept for all samples. Sections were stained with haematoxylin-eosin (H&E). The epidermis and subcutis were scored. Leucocyte infiltration, necrosis and other characteristics were assessed by a blinded certified pathologist and expressed as score with a range of 0–3: 0 was the absence of inflammatory infiltrate, 1—when the infiltrate is mild, 2—medium and 3—rich.

Al the animals were euthanised by anaesthetic overdoses after the histopathological samples were harvested.

## 3. Results

All rats considered in this study had a normal distribution of weight with a mean of 436 g and a standard deviation (SD) of 52 g for MB group and a mean of 438 g and SD of 35 g for PRO. Thus, there was no statistically relevant difference between the two groups.

All the rats included in the same group received the same amount and same concentration of dye according to the group assigned, which showed no clinical and histological differences with respect to the rat’s weight.

From the group of rats subjects infused with MB dye solution of 0.65 mg/mL, a few were excluded for various reasons as follows: one with partial flap necrosis because it was raised on two perforators, and one of them was sectioned, and the second rat due to flap autophagia in the second post-operative day; the third showed flap dehiscence, the suture wire being gnawed by the rat.

From the group of rats’ subjects infused with PRO dye solution of 1 mg/mL, two were excluded due to the formation of some island dry necrosis.

The rest of the rats’ subjects included in both groups had good clinical evolution until the moment when histopathological sample was harvested.

In the first day, it was observed that the flap was swollen, and the complete remission occurred on the second day. However, the flap started to be washed of dye from the third post-operator day, this process being related with the flap size.

### 3.1. In Vivo Study Performed on Wistar Rats after Infusion with MB Dye Solution

The images of the skin coloured with MB in the place where the functional perforasome is highlighted, with the margins of the perforasome marked, with the perforator island flap rosed and sutured in site and with the perforator island flap at 14 days after the operative procedure are presented in Figure 2a–d for a rat included in the group of rats infused with MB dye solution of 0.65 mg/mL.

The results of histological evaluation performed on various tissue samples taken from a rat in the group injected with 0.65 mg/mL of MB are shown in Figure 3.

In the first post-operatory day, after MB injection, the results of H&E staining in cross-sections, the groin artery and vein showed signs of congestion without other histopathological features. The island flap presented infiltration of inflammatory cells on dermis in medium amount (see Figure 3a). In the middle of the island where the perforator vessels ended, the quantity of inflammatory cells decreased (see Figure 3b). Then the inflammatory cells descended in hypoderm from dermis, in the same moderate quantity. Moreover, the vessels do not show any signs of lesion or inflammation. Eventually, in the fourth day, the inflammatory infiltrate started to reduce. The superficial epigastric artery and vein were without any signs of pathological changes during the post-operatory evolution. The margins of the island flap show the same infiltration of inflammatory cells on dermis, it was noticed, this time being in moderate quantity.

However, after two weeks, the flap was completely healed and integrated without signs of inflammation on subcutaneous lymph nodes (see Figure 2d).

### 3.2. In Vivo Study Performed on Wistar Rats after Infusion with PRO Dye Solution

The images of the skin coloured with PRO in the place where the perforasome is mapped, with the margins of the perforasome marked, with the perforator island flap rosed and sutured in site and with the perforator island flap at 14 days after the operative procedure presented in Figure 4a–c and Figure 5a–d for a rat included in the group of rats infused with PRO dye solution of 1 mg/mL.

In case of PRO group of rats on the first day, there were no pathological findings neither on the blood vessels nor on the skin. On the second day, one of the rats presented erosion and infiltration of mixed inflammatory cells in the inguinal artery wall. The vein presented moderate perivascular inflammatory infiltrate and erosion of the endothelium. During the evolution, it was observed that the inflammatory infiltrate reduces, while at the end of the study, after two weeks, the infiltrate is absent (Figure 6).

Clinically, all the flaps were healed and completely integrated at the end of the two weeks, (see Figure 5d).

All the histopathological findings were listed in Table 1 and were used for the statistical evaluation of the studied groups of rats.

The statistical analysis of the experimental data was assessed by using the Mann–Whitney procedure, and the U *z*-score obtained was −0.89489 while the value of *p* was 0.37346, thus being statistic significant (*p* < 0.050).

## 4. Discussion

According to the literature data [27], the assessment of flap perfusion was made in human subjects using ICG dye, to avoid partial flap necrosis [28,29]. However, it has been observed that when using this particular dye in flap perfusion, it was very difficult to differentiate the exact boundary between healthy and necrotic skin tissue [30].

Furthermore, the ability of the ICG molecules to bind on the plasma proteins makes this dye to be present only in the circulatory system [28]. Unlike ICG, MB and PRO have the property of being extracted by cells from blood, thus being a suitable dye for tissue tracer purpose [31,32,33]. The studies performed with ICG required the use of a dynamic laser-fluorescence-video angiograph, a digital video camera equipped with an infrared filter, or Fluorescence-Assisted Resection and Exploration system [34]; therefore, a complicated, difficult to handle and very expensive equipment. Moreover, when using the ICG dye, it must be dark in the examination room, so during the examination, the cautery/electric knife must be switched off to avoid artifacts. It can be thus concluded that the use of ICG dye as perforator tracer is more difficult than the direct injection of a florescent dye like MB or PRO followed by visual examination of postoperative evolution [28,35].

One of the rats from the MB group, excluded from this study, showed partial flap necrosis. The necrosis has installed because the flap was raised on two distinct perforators, one being accidentally cut. Thus, the exact edges of the skin perfused by one perforator were identified after injecting MB solution. It was demonstrated that all the excess skin, which corresponded partially to the second perforasome, necrosed [36].

To evaluate the histopathological effect of the surgical technique and of the dyes used on the tissues [26] alone, normal saline serum without dye content was injected following the same procedure. In this case, only some congestion, intraluminal thrombus or endothelial lesions, could be observed that may be due to the rigid sonde used. When the arterial dimension is bigger, a soft venous catheter could be used, avoiding arterial lesion. When injecting the dye directly intraarterially, for some of the rats, an inflammatory infiltrate was observed in the vascular wall or perivascular, and in some cases, endothelial lesions or minimal fibrosis were seen. The main advantage of the technique proposed here is the small concentration and small amount of dye needed to trace the perforator perfusion, namely, 0.65 mg/mL in the case of MB and 1 mg/mL in the case of PRO, respectively. We can thus compare our dose of only 1 mL of 0.65 mg/mL MB solution injected directly intraarterially with those reported by Ashitate et al. [30] that used 2.0 mg/kg of MB dye to evaluate the flap perfusion in pigs. In our case, the amount of MB dye used is significantly smaller, but the arterial pick was registered at 5 min after injecting the dye in the external jugular vein.

Stradling et al. [37] found that 21% of the patients included in their study developed skin lesions after 5 mL of 1% MB solution were injected. Superficial ulceration, intense erythematous lesion, or a necrotic lesion developed at the site of injection. In our study, the site of injection was the femoral artery, and it was observed that some inflammatory infiltrate was registered, even when only 0.9% saline was injected following the same procedure. Thus, it was considered that this effect is due to the injection manoeuvres performed and not to the dyes used.

When the dye was injected in the superficial dermis, adverse reaction was registered on the skin, like intense erythematous macular lesions [38], superficial ulcers with deep pallor and necrotic ulcerations which may correspond to the histopathological findings. We observed that the inflammatory infiltrate in the skin island is absent in lower concentrations of dye (0.325 mg/mL) for MB respective to 0.5 mg/mL PRO, but the tracer is still visible without the need for any special condition or technology. Therefore, this favourable observation can be further exploited for sentinel nodule identification using a lower concentration of dye that could reduce, even preventing adverse effects on skin when the dye is injected in breasts.

When the proper concentration was evaluated to identify the perforasome, it was observed that in the case of MB, 0.65 mg/mL was the optimal one, and the total volume used was of 1ml. In this case, the inflammatory infiltrate was observed in the first day but only in moderate quantity, then being absorbed until the 14th day after the surgical procedure. On the other hand, in the case of PRO, the proper concentration 1 mg/mL and a volume of 1 mL were the optimal since it showed rich or moderate inflammatory infiltrate. Lower concentration as 0.5 mg/mL shows no signs of inflammatory infiltrate but more than 1ml is usually required to identify the entire perforasome. When more than 1 mL of MB is injected, choke vessels close due to the action of nitric oxide synthase (NOS) [39]. MB inhibits the NOS decreasing the quantity of NO [40]. When more than 1 mL MB was injected, the first surface coloured was reduced in time, and other territories coloured then later. This demonstrates the MB effect on choke vessels by closing one territory and opening another. On the other hand, MB reduces inflammation by acting on NOS, thus reducing the quantity of highly reactive peroxynitrite (ONOO^−^) [41,42]. Moreover, the histopathological evaluation of the skin island in time revealed the decrease of the extensive damage to cellular lipid components [43,44]. After 14 days from the surgical procedure, no infiltrate was observed. In the same study, it was reported that the systemic circulation increased via NOS inhibition which is a huge advantage for the flap [43].

In addition, MB acts as preconditioning pharmaceutical agent for flap irrigation by closing the choke vessels before the flap elevation begins [45].

PRO has the advantage of being highly fluorescent [46], which can be seen directly with the naked eye, on a relatively low concentration (1 mg/mL), without any vasoconstrictor effect on choke vessels like the one produced by MB injection [47].

Even though all the rats received the same amount of dye, 1 mL of solution having the same concentration of dye being injected to each subject according to the group assigned, no clinical and histological differences were observed depending on rat’s weight. This is probably due to most of the dye remaining in the island and being washed out gradually.

MB seem to be a better choice than ICG to evaluate the functional perforasome; further studies need to be done in order to establish if a PRO concentration of 0.1mg/mL will have the same efficiency as in the case of MB in order to determine the functional perforasome without the presence of the inflammatory infiltrate in the skin island.

Furthermore, a better solution should be found for the injection of the dye solution since the use of metallic canula can cause endothelial lesions and inflammatory infiltrate on vascular wall.

For human use, the amount of dye needed should be adjusted to obtain the expected result.

## 5. Conclusions

In conclusion, both medical dyes used in this study proved to be suitable for the in vivo use as functional perforator tissue tracer. Although some local tissue reactions were registered, finally the flaps healed and integrated perfectly for all subjects considered in the study, regardless of which group they belonged to. Injecting the tracer dye directly intraarterially has several advantages like the reduction of the time required for evaluation. It also avoids injecting a large amount while there is concentration of dye into the circulatory system to allow direct observation, without any impediment, of the surface of the functional perforasome.

Although the results obtained so far are promising, further studies are needed to validate this method for use on human subjects in clinical trials.

## Figures and Tables

**Figure 1 jpm-11-00147-f001:**
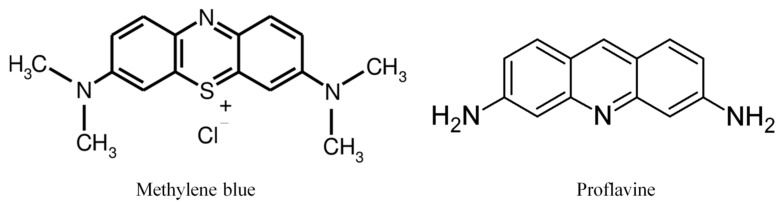
The chemical structure of methylene blue (**left**) and proflavine (**right**).

**Figure 2 jpm-11-00147-f002:**
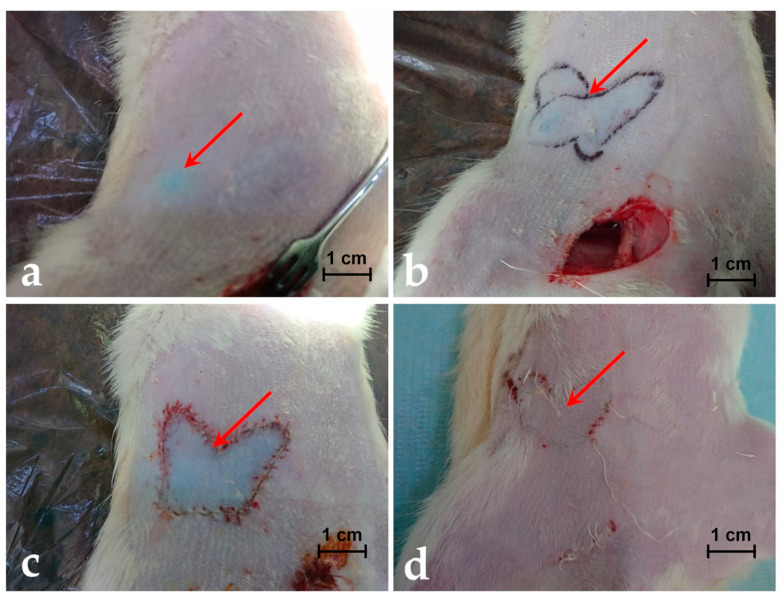
(**a**) Image of the skin coloured with MB in the place where the functional perforasome is highlighted. (**b**) Image with the margins/edges of the perforasome marked for better view. (**c**) Image with the perforator island flap rosed and sutured in site. (**d**) The perforator island flap at 14 days after the operative procedure.

**Figure 3 jpm-11-00147-f003:**
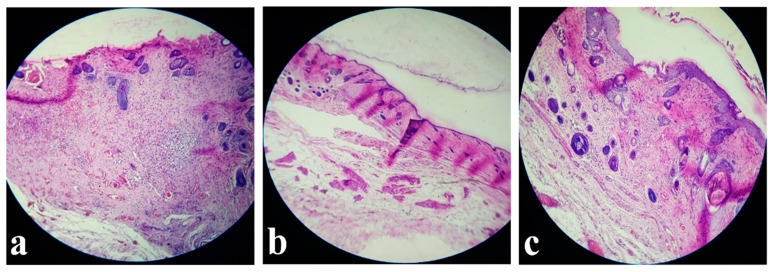
Histological images illustrating: (**a**) Island margins/edges with inflammatory infiltrate; (**b**) middle of the island without histopathological changes; (**c**) minimal inflammatory infiltrate in dermis and vascular congestion (10×).

**Figure 4 jpm-11-00147-f004:**
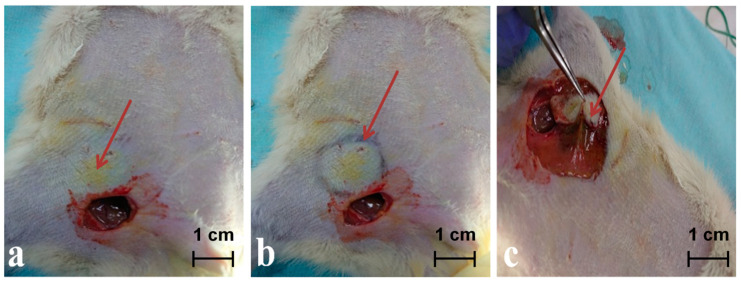
(**a**) Image of the functional perforasome marked by PRO intraoperatory. (**b**) Image with the margins/edges of the perforasome marked for better view. (**c**) Identification of coloured perforator.

**Figure 5 jpm-11-00147-f005:**
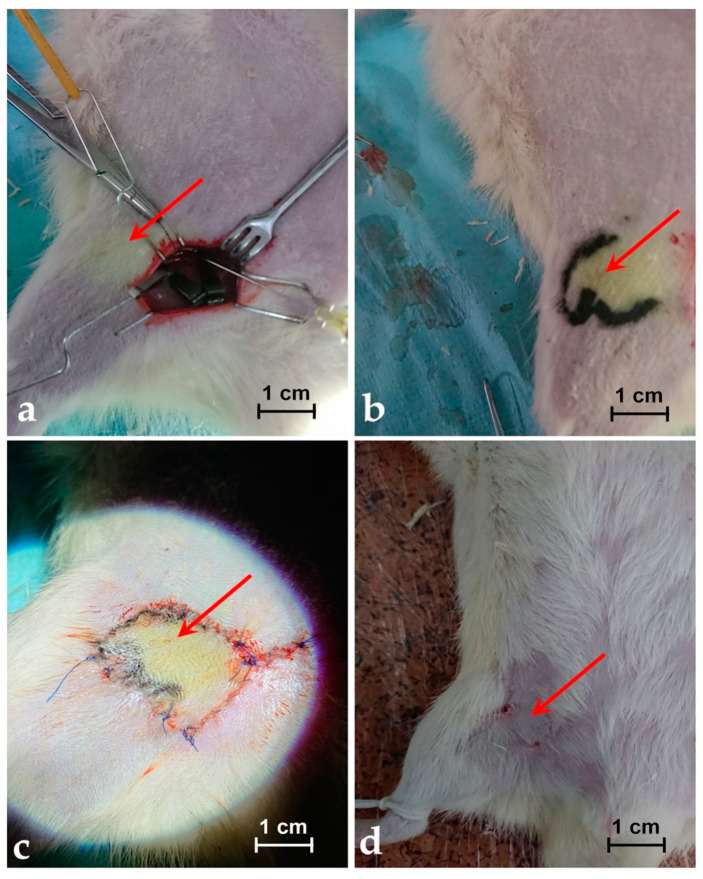
(**a**) Image of the skin zone coloured by PRO. (**b**) Image with the margins/edges of the functional perforasome marked for better view. (**c**) Image with the skin perforasome island after rising and suturing. (**d**) Image with the flap integrated on the 14th day after operatory procedure.

**Figure 6 jpm-11-00147-f006:**
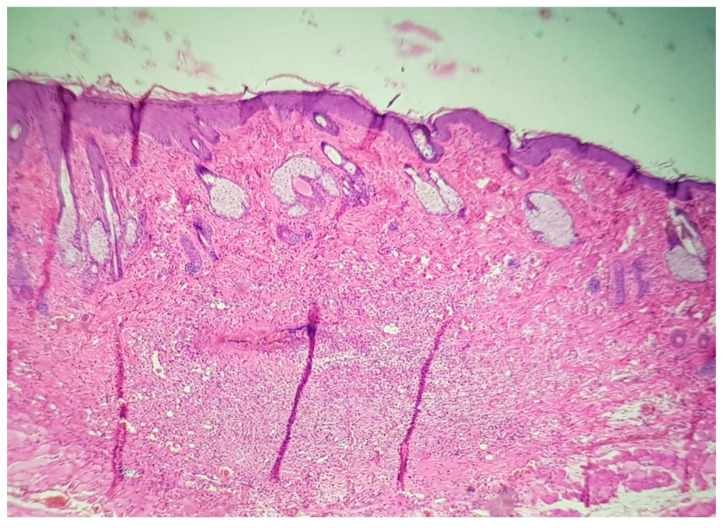
Histological image illustrating skin with granulate tissue in dermis on the 14th day after operatory procedure (20×).

**Table 1 jpm-11-00147-t001:** Quantity of mixed inflammatory infiltrate in skin island after dye injection.

Batch Number	Rat Number	Evaluation of the Inflammatory Infiltrate	
		Injection of 0.65 mg/mL MB	Injection of 1 mg/mL PRO
Batch 1	1	Medium	Medium
	2	Medium	Rich
	3	Medium	Medium
	4	Medium	Rich
	5	Absent	Absent
	6	Absent	Absent
Batch 2	1	Medium	Medium
	2	Medium	Rich
	3	Medium	Medium
	4	Medium	Rich
	5	Absent	Absent
	6	Absent	Absent

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
