# Peer review of "Methylene Blue and Proflavine as Intraarterial Marker for Functional Perforazome—Comparative Study"

_jpm, 2021, doi:10.3390/jpm11020147_

Round 1

Reviewer 1 Report

The manuscript “Methylene blue and Proflavine as Intraarterial Marker for Functional Perforazome - Comparative Study” (Nedu et al.) reported the comparison of two dyes (i.e. MB and PRO) as perforator tracers.

The following observations have been raised during the reviewing:

  • For sake of clarity, chemical structures of MB and PRO should be added
  • Authors should comments on the rationale behind the selection of the concentration of MB and PRO used in the experiments.

Author Response

Answer for Reviewer 1:

The manuscript “Methylene blue and Proflavine as Intraarterial Marker for Functional Perforazome - Comparative Study” (Nedu et al.) reported the comparison of two dyes (i.e. MB and PRO) as perforator tracers.

Answer: We thank the reviewer for carefully reading our manuscript and for the observations that allow us to increase its quality. We have answered to all observations and we hope that our response will clarify all the issues highlighted by the reviewer. The manuscript was revised and completed according to the recommendations, all changes being highlighted in yellow.

The following observations have been raised during the reviewing:

  • For sake of clarity, chemical structures of MB and PRO should be added

Answer: Chemical structures for methylene blue and proflavine were added in new Figure 1, as it can be seen in the revised manuscript.

  • Authors should comments on the rationale behind the selection of the concentration of MB and PRO used in the experiments.

Answer: The concentrations for methylene blue and proflavine used in this study were selected based on the preliminary results obtained after the injection of dye solutions of different concentration into rats. After the study of literature (see reference [31] in the manuscript), a series of solutions with concentrations between 0.16 - 1.6 mg/ml for MB (see reference [26] in the manuscript), and 0.1-1 mg/ml for PRO, respectively, were prepared and injected into rats. The lowest concentration that allowed the easy visualization with naked eye after injection was chosen for each dye.

The following paragraph was introduced in the Materials and methods section of the manuscript:

 “A series of solutions with concentrations between 0.16-1.6 mg/ml for MB, and 0.1-1 mg/ml for PRO, respectively, were prepared and injected into rats. The lowest concentration which allowed easy observation with the naked eye after injection was chosen for each dye.”

 Reviewer 2 Report

The manuscripts considers comparision between two common dyes: Methylene blue and Proflavine as functional perforasome tracers.
Thie subject is important especially in microsurgical free flap transfer. Quality of the dye is a crucial factor in skin perfusion assessment.

This paper needs major improvement especially in presentation of results. The images are poor quality. Fig. 1a&b are photos of the screen which results in nonacceptable Moire effect. In my opinion also the "Leica" logo should not be presented. Overall, the images with the rat body presented are to dimm and notproperly composed, they should be cropped for instance Fig. 4c. Fig. 5 is blurry and distorted. 
Please add scale to the images.

The description of inflammatory infiltrate evaluation is poor. I strongly suggest for the future version of the paper to include some quntitative method and add its result in Table 1.

Author Response

Answer for Reviewer 2:

The manuscripts considers comparision between two common dyes: Methylene blue and Proflavine as functional perforasome tracers.
Thie subject is important especially in microsurgical free flap transfer. Quality of the dye is a crucial factor in skin perfusion assessment.

Answer: We thank the reviewer for the favorable comments and appreciations on our work. The observations allowed us to increase its quality of the manuscript. We have answered to all observations and we hope that our response will clarify all the issues highlighted by the reviewer. The manuscript was revised and completed according to the recommendations, all changes being highlighted in yellow.

This paper needs major improvement especially in presentation of results. The images are poor quality. Fig. 1a&b are photos of the screen which results in nonacceptable Moire effect. In my opinion also the "Leica" logo should not be presented. Overall, the images with the rat body presented are to dimm and not properly composed, they should be cropped for instance Fig. 4c. Fig. 5 is blurry and distorted. 
Please add scale to the images.

Answer: We thank the reviewer for these suggestions. All the mentioned images were revised as suggested by reviewer, and their quality was improved. All the changes made to the figures or corrections in the text can be seen in the revised version of the manuscript.

The description of inflammatory infiltrate evaluation is poor. I strongly suggest for the future version of the paper to include some quntitative method and add its result in Table 1.

Answer:  The evaluation of the viability of the tissue taken from the flap by means of the histopathological examination was the optimal method, but in this case there is no standardized version for quantitative evaluation. The indicators used in Table 1 for the evaluation of the inflammatory infiltrate are those commonly used in practice.

Round 2

Reviewer 2 Report

All my suggestions have been revised. I still recommend professional English editing before publication.